# Language and Hidden Emotion Understanding in Deaf and Hard-of-Hearing Children: The Role of Mentalistic Verbs

**DOI:** 10.3390/bs15081106

**Published:** 2025-08-15

**Authors:** Alaitz Intxaustegi, Elisabet Serrat, Anna Amadó, Francesc Sidera

**Affiliations:** 1Department of Psychology, Universitat de Girona, 17004 Girona, Spain; elisabet.serrat@udg.edu (E.S.); francesc.sidera@udg.edu (F.S.); 2Department of Psychology, Sociology and Social Work, Universitat de Lleida, 25001 Lleida, Spain; anna.amado@udl.cat

**Keywords:** social cognition, belief attribution, mental state reasoning, childhood deafness, language development

## Abstract

The understanding of hidden emotions—situations in which individuals deliberately express an emotion different from what they genuinely feel—is a key skill in theory of mind (ToM) development. This ability allows children to reason about discrepancies between internal emotional states and external expressions and is closely tied to linguistic development, particularly vocabulary related to mental states, which supports complex emotional reasoning. Children who are deaf or hard of hearing (DHH), especially those born to hearing non-signing families and raised in oral language environments, may face challenges in early language exposure. This can impact the development of social and emotional skills, including the ability to understand hidden emotions. This study compares the understanding of hidden emotions in hearing children (*n* = 59) and DHH children (*n* = 44) aged 7–12 years. All children were educated in spoken language environments; none of the DHH participants had native exposure to sign language. Participants completed a hidden emotions task involving illustrated stories where a character showed a certain emotion in front of two observers, only one of whom was aware of the character’s true emotional state. The task assessed children’s understanding of the character’s emotional state as well as their ability to reason about the impact of hiding emotions on the beliefs of the observers. The results showed that the hearing children outperformed their DHH peers in understanding hidden emotions. This difference was not attributed to hearing status per se but to language use. Specifically, children’s spontaneous use of cognitive verbs (e.g., think or know) in their explanations predicted task performance across the groups, emphasizing the role of mental state language in emotional reasoning. These findings underscore the importance of early and accessible language exposure in supporting the emotional and social cognitive development of DHH children.

## 1. Introduction

[1] ([1]) described theory of mind (ToM) as the ability to understand that people have mental states such as thoughts, beliefs, desires, intentions, and emotions that influence their behavior. This means recognizing that others may think or feel differently from us and using that understanding to interpret or predict how they might act. Therefore, ToM is a crucial skill for social interaction, communication, empathy, and learning. A commonly accepted distinction in the ToM literature differentiates between its cognitive and affective components ([26]; [9]). Cognitive ToM refers to the capacity to infer others’ beliefs, intentions, or thoughts, whereas affective ToM involves understanding and reasoning about others’ emotional states. In this manuscript, we focus on a particular manifestation of ToM competence: the understanding of hidden emotions, in which individuals deliberately conceal their true feelings. These scenarios are especially relevant because they require the integration of both affective and cognitive ToM. Specifically, understanding hidden emotions entails recognizing an inner feeling that differs from someone’s expression (affective ToM) but also understanding that this faked expression may influence people’s beliefs about the feelings of this person (cognitive ToM) ([27]).

In the following section, we briefly review how hearing children develop the ability to understand hidden emotions before examining how this process may differ in deaf and hard-of-hearing (DHH) children.

### 1.1. Understanding Hidden Emotion

Emotional understanding is a gradual process from infancy to adulthood. Infants begin responding to emotions like happiness, surprise, anger, and fear within the first few months, and by 12 months, they use others’ emotional cues to guide their behavior. By 18 months, they can infer others’ desires, even if these differ from their own. By age two, children begin to understand and regulate emotions, a development that continues into early adolescence ([37]). With the emergence of language, during the first two years of life, children start to label basic emotions (happiness, sadness, etc.) based on facial expressions, movements, and voices ([36]). During their second and third years of life, children start to understand which situations cause certain emotions and that people’s emotions depend on their desires. Around the age of four, children start to understand false beliefs (that people’s representations from reality may differ from reality, such as a child believing he has a sweet in his pocket, but in reality, he lost it) ([36]). From the age of four, most children understand that people may pretend or simulate emotions ([44]), although they still struggle to differentiate explicitly that in these situations, people’s external feelings may be different from their internal feelings. This knowledge appears between the ages of 4 and 6 years ([22]; [55]) and reflects an important milestone in emotional development, as it implies an understanding that individuals may internally experience emotions that differ from those they outwardly express. This understanding can be classified as first-order affective ToM, because it involves understanding people’s feelings ([51]).

Once children differentiate inner from external emotions, they are able to start to reason that hiding emotions may affect the beliefs of the observers of these emotions. Prior research suggests that at the age of six, children still struggle to understand this kind of belief, despite understanding that emotions may be hidden ([22]). This higher difficulty may be explained by it involving a higher degree of reasoning, namely second-order reasoning about mental states, which involves understanding that a person may have a false belief about the mental state (an emotion) of someone else. The study by [47] ([47]) suggested that children develop such an understanding mostly between the ages of 6 and 8 years. However, they used tasks of children hiding emotions where, in order to respond correctly to the beliefs of the observers, the children could succeed by relying solely on the observer’s facial expression without needing to consider that the protagonist was hiding an emotion, and thus their results could not be entirely conclusive. Hence, in the present study, we designed a task with a key methodological improvement compared with theirs; our task included two observers of the child’s emotional simulation, namely one character who was unaware of the child’s real feelings in that situation and another who was aware because they had witnessed the event. This change in the design of the task should advance prior knowledge about children’s capacity to understand the consequences of hiding emotions for other people’s beliefs by overcoming prior methodological problems.

### 1.2. Hidden Emotion Understanding and Language

In the development of emotional understanding, language plays a crucial role in representing and organizing emotional concepts and meanings. It supports emotional exchanges and enables children to connect specific expressions, actions, or events with their corresponding external referents while also helping label the emotions they elicit. Research has shown that the temporal synchrony between vocal intonation, facial expressions, spoken language, and emotional stimuli, such as objects or events, is essential for young infants to learn and discriminate between emotional states ([13]).

Most of the studies on the relationship between ToM and linguistic skills have focused on studying the linguistic influences on first-order ToM, such as in situations involving an understanding of people’s false beliefs. This research has found that various linguistic skills are important for the development of ToM, such as vocabulary ([40]), mental state vocabulary ([28]), conversational experiences ([28]), epistemic verbs ([39]), or the general understanding of language ([5]).

Some other studies have also focused on studying which specific linguistic skills are important for being able to reason about simulated emotions. In this respect, [44] ([44]) found that understanding pretend emotions (understanding emotions simulated in the context of pretend play) was linked to vocabulary but especially to the syntax of complementation, with both communicative verbs (as in “Maria said that…”) and with cognitive verbs (as in “Maria thought that…”). In another study, [45] ([45]) found that DHH children’s delayed understanding of pretend emotions was strongly associated with their difficulties in expressive vocabulary and pragmatics.

With reference to semantic-lexical development, the availability of terms and their associated concepts to describe one’s own and others’ mental states is essential for the initial acquisition of ToM and for overcoming false belief tasks ([3]). Between the ages of 2 and 4, children begin to use mentalistic language first as a conversational tool and then to refer to mental states ([2]; [15]; [43]), first in reference to themselves and later to others ([16]).

Around 2 years of age, children use mentalistic language related to basic desires, perceptions, and emotions, evolving toward more complex concepts such as thoughts and beliefs from the age of 3 years ([11]; [28]). Aside from this, in children aged 8–10, mentalistic and metacognitive language use has been found to predict performance in emotional comprehension and false belief tasks ([15]). Mentalistic terms referring to desires and beliefs imply similar intentional states but do not appear simultaneously in children’s language ([29]). Verbs of desire (e.g., “I want”) emerge around age two, while verbs of belief (e.g., “I know” or “I believe”) appear around age three ([3]; [12]; [23]). Thus, the comprehension of volitional verbs (e.g., “I want”) precedes that of cognitive verbs (e.g., “I think” or “I know”), both due to their level of complexity and their development in children’s spontaneous language. This sequence reflects an evolutionary pattern consistent with the acquisition of mentalistic language and ToM, where desire precedes belief as a form of attribution of mental states ([25]).

### 1.3. Emotion Comprehension in DHH Children

Thus far, we have reviewed emotion understanding skills (and their relation with language components) in hearing children. We will now focus on DHH children. Several investigations suggest that DHH children born to hearing (non-signing) parents (DoH children) may show differences in ToM development, compared with their hearing peers, in understanding the mental states of others, such as desires, beliefs, or intentions, and emotions in particular (e.g., [35]; [40]). Therefore, the emotional understanding skills of deaf children must be framed in more general skills that they have when it comes to understanding people’s mental states and behaviors.

There is also evidence that language plays a role in the development of ToM skills in DHH children ([24]; [42]; [48]; [53]), who sometimes show poorer social and emotional knowledge skills than hearing children. In this regard, however, some researchers have suggested that the ability of parents to communicate with children through sign language from birth, as is the case with deaf children born to deaf parents (DoD children), is crucial for the development of socio-cognitive skills ([14]; [19]; [38]). There is long evidence nowadays that DoD children outperform DoH children (either oral communicators or late signers) in ToM tasks and that DoD children usually perform similarly to hearing peers of the same age (see [34]). Early access to a first language—whether spoken or signed—is essential for the development of children’s understanding of mental and emotional states. Equally important is the opportunity to use that language to engage in conversations about others’ thoughts and feelings at home and at school ([19]).

DoH children often experience delays in language acquisition, which can in turn hinder the development of social and emotional skills such as the understanding of hidden or simulated emotions ([48]; [53]; [54]). Limited access to language rich in emotional and cognitive nuances restricts the acquisition of crucial vocabulary, such as pretense verbs, which are fundamental for interpreting feigned emotions ([7]). Recent evidence reinforces this view, as [6] ([6]) showed that DHH children with stronger language comprehension performed better in ToM tasks involving discrepancies between internal emotional states and outward expressions. In line with this, the study conducted by [18] ([18]) with deaf children who received cochlear implants showed that early implant children (within 18 months of age) had better emotion comprehension skills than those who received implants later. In fact, different linguistic and communicative skills have been found to be related to DHH children’s abilities to understand simulated emotions, especially those related to the use of language ([45]). Understanding hidden emotions may pose a particular challenge for DHH children not only because it requires dual-level reasoning (involving both affective and cognitive ToM) ([22]; [51]) but also because it relies on access to nuanced emotion language, such as cognitive verbs and pragmatic markers, which are often less available in environments with delayed or limited language exposure ([6]; [7]; [40]; [45]).

In the present study, we examine how children interpret scenarios where emotional expressions differ from internal feelings. Specifically, we compare the performance of DHH children and hearing children in a task that requires understanding both emotional concealment and its impact on others’ beliefs. In the process, we also consider the language used in their responses, particularly the presence of mental state terms such as cognitive verbs. In this context, the use of mental state verbs has been examined in relation to complex morphosyntactic structures that influence their production. Accordingly, [32] ([32]) found that children who are DHH tend to use more perception-based syntactic structures rather than cognition-based ones. Moreover, in a study with 4-year-old children, [49] ([49]) identified differences in the use of mental state verbs between DHH and hearing children. Specifically, their findings showed that children who are DHH use significantly fewer mental state verbs in complex syntactic constructions compared with hearing children. Furthermore, they used a smaller variety of mental state verbs and significantly fewer of these verbs in obligatory contexts.

Based on previous research highlighting the link between language development and theory of mind, we expect that hearing children may show higher levels of accuracy in a task on hidden emotions than DHH children, and the use of mentalistic language will be associated with better performance. This study aims to contribute to a better understanding of how language exposure and communicative experience relate to children’s ability to interpret emotionally complex social situations.

## 2. Materials and Methods

### 2.1. Participants

The study included a total of 103 children aged between 7 and 12 years who were divided into two groups: children with hearing loss (DHH) and children with typical hearing (see Table 1).

The 44 children of the DHH group were all diagnosed with permanent bilateral hearing loss by a certified audiologist and did not use sign language. All participants (except one) in this group used assistive hearing technology. Specifically, 23 children used bilateral hearing aids, 15 had bilateral cochlear implants (CIs), 2 used a single CI, and 3 used a combination of one CI and one hearing aid. All DHH participants received support from language therapists in their mainstream schools.

The control group comprised 59 hearing children. Statistical analyses through a Mann–Whitney U test revealed no significant differences between the two groups in terms of age (*U* = 1159; *p* = 0.33) or non-verbal IQ (*U* = 1143.5; *p* = 0.30). Furthermore, a chi-squared test showed that the gender distribution did not differ significantly by group (*χ*^2^ = 0.068; *p* = 0.79); the proportions of boys and girls in each group were quite similar.

### 2.2. Materials

Each participant completed the following tasks individually in the sequence detailed below.

### 2.3. Non-Verbal Intelligence

To assess and confirm that the non-verbal intelligence levels were equivalent between the groups, the matrices subtest from the Kaufman Brief Intelligence Test (K-BIT; [17]) was administered. This subtest evaluates the ability to identify relationships among visual stimuli by selecting the correct option from a set of multiple-choice responses. Standard scores were then derived based on each participant’s performance.

### 2.4. Hidden Emotions Task

To assess the comprehension of hidden emotions, we used an adapted version of the hidden emotion deception task designed by [46] ([46]) (see Appendix A). The story was illustrated with two color drawings shown sequentially to each child to accompany the narrative. The children were first asked to identify both the external and internal emotions of the story’s protagonist and then answer questions regarding the beliefs of the observers (the parents) in the scenario. Justifications for the participants’ responses about the protagonist’s emotions (both external and internal), and the observers’ beliefs were recorded. The task also included three control questions to verify comprehension of the story. The majority of the children (86.4%) responded correctly to all memory questions while 11.7% failed one question, and only two children (one from each group) failed two questions. However, we did not exclude them for their responses to the memory questions, as we prioritized their performance in understanding the emotions and beliefs of the characters.

The scoring of the task was as follows. Children were awarded 1 point for correctly answering both the external and internal emotion questions. A second point was given for accurately identifying the beliefs of the observers, specifically selecting the protagonist’s internal emotion for the parent who witnessed the previous situation and the protagonist’s external emotion for the parent who did not witness it. Finally, a third point was awarded when the child provided a clear and appropriate justification for their response to why the protagonist showed the external emotion. We considered a justification about the external emotion correct if it referred to the fact that the child was lying or trying to deceive her parents.

Aside from these scores, we recorded whether the children, in this justification, used a volitional verb (e.g., want) or a cognitive verb (e.g., think, believe, know, or pretend). This distinction was based on the semantic classification by [4] ([4]), who defined the following categories for internal state language: perception, physiology, affect, moral judgement and obligation, cognition, and volition and ability. Furthermore, constructions such as “*wants to make someone believe*” were coded as cognitive rather than volitional, as they refer to mental states. Similarly, the verb “*lie*” was classified as cognitive; although not strictly so, its use was considered to imply taking the other person’s beliefs into account.

### 2.5. Recalling Senteces

We administered the sentence repetition subtest from *Clinical Evaluation of Language Fundamentals—Fifth Edition* (CELF-V; [52]), a standardized instrument developed to evaluate language skills in children aged 5–15 years. Despite the manual of the CELF-V not providing a validation of the test for DHH children, in the study by [8] ([8]), where they evaluated hearing and deaf children with CELF-V, they found that this test showed high internal consistency in the deaf group for the different subtests administered to the children, including the recalling sentences subtest.

This subtest assesses a child’s capacity to accurately reproduce sentences that progressively increase in length and syntactic complexity while preserving their original meaning, structure, and content. The subtest consists of 26 items, with different starting points depending on the child’s age, and responses are scored on a scale from 0 to 3 based on the number and type of errors.

In addition to the tasks administered to the children, a brief questionnaire was completed by either their parents or speech therapists to collect relevant sociodemographic data (for both DHH and hearing children: gender, date of birth, age at school entry, languages used by the parents to communicate with the child, language spoken between the parents, learning difficulties, and medical conditions) and hearing information (for DHH children only: age of onset and cause of hearing loss, age at diagnosis, degree of hearing loss in each ear, type and age of onset of audiological devices, and relatives who are deaf or hard of hearing).

### 2.6. Procedure

Ethical approval for the study was granted by the ethics committee of the institution affiliated with the first author, and informed consent was obtained from the parents of all participants.

DHH children were recruited through public services that provide support to children with hearing or language difficulties. Within these services, speech therapists conduct regular weekly visits to schools, and it was through this network that families and schools were initially contacted. Speech therapists were present during the assessment sessions in an observational capacity, without directly engaging in the evaluation procedures. Children with typical hearing were recruited from public schools. All tasks were administered over two separate sessions, each lasting approximately between 25 and 40 min, and they took place in a quiet room within the child’s school. The sessions were conducted by researchers with prior experience administering language and ToM tasks to both DHH and hearing children.

Data analysis was conducted using IBM SPSS Statistics version 29.0.2.0. First, descriptive and correlational analyses (Spearman’s correlation coefficients) were performed to summarize the scores of the study variables and explore the relationships among them. Subsequently, chi-squared and Mann–Whitney U tests were applied to compare group performance for the hidden emotion understanding task, considering both the score for correct responses and the combined score including responses and justifications. A chi-squared test was also conducted to examine potential qualitative differences between groups in the use of cognitive and volitional verbs in participants’ justifications. Finally, a multiple linear regression analysis was performed to identify predictors of task performance, including linguistic, cognitive, and age-related variables and group membership.

## 3. Results

Distributional checks indicated non-normality and variance heterogeneity for some variables; thus, non-parametric tests were used where appropriate.

Table 2 presents the descriptive statistics for the variables used in the analysis, which are split according to group. The hearing children showed higher means for all measures, particularly language ability (RS_CELF) and the use of cognitive verbs.

Given the non-normal distribution, a non-parametric comparison of group means (Mann–Whitney U test) was carried out to examine differences in performance for the hidden emotions comprehension task. The results show that the children in the control group performed significantly better than the DHH children. This significant difference remained consistent whether only the correctness of the responses was considered (score; *U* = 1047.50; *p* = 0.005) or both the responses and their justifications (justification; *U* = 859.50; *p* = 0.001), with a more significant difference in the latter.

To further examine the group differences in response accuracy, Table 3 presents the proportions of correct responses to the emotion and belief questions separately in the hidden emotion task. As shown in Table 3, the hearing children showed better accuracy across both types, with p values close to significance level.

The following Spearman’s correlation coefficients were computed among the main study variables (see Figure 1): language ability (recalling sentences from CELF), use of volitional verbs, use of cognitive verbs, performance for the hidden emotion understanding task, and a composite measure combining task performance and justification quality. Significant correlations were found between performance for the hidden emotion task and language ability (*p* < 0.001) and between answer justification and language ability (*p* < 0.001). The use of cognitive verbs also correlated significantly with both performance (*p* < 0.01) and justification (*p* < 0.001).

To investigate in more depth the differences in the qualitative nature of the participants’ responses between the two groups, a chi-squared analysis was conducted comparing the types of verbs used in the response justifications (cognitive and volitional). No significant differences were found between groups in the use of volitional verbs. However, a significant difference emerged in the use of cognitive verbs: the children in the control group used these verbs more frequently than their DHH peers when explaining their answers (See Table 4).

A multiple linear regression analysis (Table 5) was conducted to examine the predictors of performance in the hidden emotion understanding task (HE_Score). Prior to performing the regression analysis, we assessed the assumptions of linearity, independence, homoscedasticity, and normality of the residuals using scatterplots and Q–Q plots. No significant violations were observed. The model included the following continuous predictors: language ability (RS_CELF), age in months, and the use of two types of verbs, namely cognitive (C_Verbs) and volitional (V_Verbs). Group (DHH vs. control) was included as a categorical predictor, with the control group as the reference category.

The results showed that the use of cognitive verbs was a significant positive predictor of performance (β = 0.255, *p* = 0.004). Additionally, age emerged as a significant predictor (β = 0.255, *p* = 0.002), indicating that older children tended to perform better at the task. Recalling sentences (CELF) also showed an effect (β = 0.232, *p* = 0.017), pointing to a potential contribution of broader linguistic skills. In contrast, nonverbal reasoning and the use of volitional verbs were not significant predictors, indicating that these factors may be less relevant in explaining performance for this task. Finally, no significant differences were found between groups (β = 0.044, *p* = 0.637). The model showed moderate explanatory power, with *R*^2^ = 0.226 and the adjusted *R*^2^ = 0.197.

## 4. Discussion

This study aimed to investigate how non-signing DHH children understand hidden emotions compared to their hearing peers and explore the role of language in contributing to this socio-cognitive ability. As expected, and in line with [35] ([35]), the hearing children significantly outperformed the DHH children at identifying both internal and external emotional states, as well as justifying their answers in a hidden emotion task. Understanding hidden emotions involves both affective ToM—grasping the distinction between felt and expressed emotions—and cognitive ToM, or inferring how an observer may misinterpret those emotions based on limited knowledge. Previous studies ([28]; [44], [45]; [49]) have highlighted the role of syntactic and lexical competencies, especially the use of mental state verbs, in enabling children to engage in this type of reasoning. The current findings extend this evidence by showing that hearing children were more likely to use cognitive verbs (e.g., “pretend,” “think”, or “believe”) in their justifications than their DHH peers. This linguistic resource seems to be essential for articulating second-order beliefs and managing emotionally complex social scenarios.

Interestingly, while the DHH children demonstrated lower performance overall, the regression model revealed that group membership was not a significant predictor when language abilities and age were accounted for. This supports the notion that DHH children’s challenges in ToM may not be due to auditory deficits per se but rather limitations in linguistic input and conversational exposure, particularly in families where sign language is not used (see [20]; [24]; [35]). The predictive power of both age and cognitive verb usage reinforces the idea that both developmental maturity and language are key to performing hidden emotion tasks. In this regard, we need to highlight that our results do not apply to deaf signers, namely DoD children who have sign language as their native language, as the literature suggests that unless they have a limitation on the use of their sign language in their environment, their performance in ToM tasks is similar to that of their hearing peers ([34]).

The present study also builds on previous research by improving the design of the hidden emotion task. Whereas previous tasks (e.g., [21]; [47]) could be solved based solely on observers’ facial expressions, our version introduces two observers with different access to contextual knowledge. This manipulation forces children to engage in second-order reasoning; they must understand not only that emotions can be hidden but also that observers can form false beliefs about those emotions based on what they have or have not seen. This variation allows us to assess adequately the understanding of the beliefs of the observers in hidden emotion situations, which will be helpful for research on this topic in future studies.

Our manuscript confirms that, in line with what prior research suggested (e.g., [47]), hearing children from the age of eight have little difficulty with understanding that hiding emotions may mislead observers. In addition, our results show that some non-signing DHH children still have trouble with understanding situations where external and internal emotions differ in the same way that other studies have found for children under the age of eight ([6]; [45]). In [6] ([6]) study with 5-year-old children, the hidden emotions subscale was the only ToM subscale in which children with mild-to-moderate hearing loss showed significant differences compared with their peers without hearing loss. This indicates that it is a task that poses particular difficulties for DHH children. Our results are interesting because they focus on an area of socio-cognitive development that has been understudied in recent years (whether children realize that hiding emotions may have consequences on the belief of certain observers as a function of the knowledge of the observers about the situation), and by showing that some DHH children may have difficulty with this skill, educators should make sure that children can reason adequately in this kind of situation and help them develop and apply this knowledge. One approach for this could be using stories about hidden emotions in which the beliefs of the stories are depicted as thought bubbles, as [50] ([50]) found that deaf children’s ToM understanding can improve by having conversations about false beliefs using thought bubbles and mental state terms.

Furthermore, our findings align with the developmental trajectory proposed in the literature, from basic emotional labelling in early childhood ([36]) to more complex reasoning involving desires, beliefs, and simulated emotions ([3]; [28]). According to [15] ([15]), between the ages of 8 and 12, children are expected to rely on metacognitive and mentalistic language to interpret and communicate about others’ internal experiences. However, the DHH children in our sample—who had delayed access to language-rich environments—seemed to lag in this progression, particularly in the use of mental state language. The distinction between volitional and cognitive verbs observed in our analysis further supports this point. While no significant group differences were found in the use of volitional verbs, cognitive verbs were both less frequently used and strongly associated with better performance in the hidden emotions task. These results are in line with those obtained by [49] ([49]) in a study with DHH children under the age of five, in which they found that DHH children use fewer mental state verbs in complex syntactic constructions compared with hearing children. This, together with the fact that hearing parents of profoundly deaf children generally use less mental state talk (compared with hearing parents of hearing children) ([24]), highlights the importance of supporting the conversations about mental states in DHH children ([31]). This also aligns with prior research indicating that verbs such as “want” and “like” emerge earlier in development, whereas more complex cognitive verbs like “believe” or “pretend” appear later and require more advanced language skills ([3]; [29]). The limited use of cognitive terms observed in DHH non-signing children in our study when giving explanations about the emotional expression of the character when he was hiding his real emotion could be perhaps fostered by having cognition-oriented conversations that include mental state terms (like think, know, intend, or pretend) and where children are asked about the thoughts and knowledge of the characters and themselves ([33]).

In sum, the findings of our study emphasize the crucial role of language, particularly mental state vocabulary, in the development of emotional understanding and ToM. The results suggest the need to foster rich linguistic environments for DHH children in both school and family settings to support their socio-cognitive development.

### Limitations and Future Studies

One main limitation of this study lies in the use of a structured, one-time task to assess the understanding of hidden emotions, which may restrict the generalizability of the results to more natural social settings. Future research could complement this approach with ecological observations or reports from significant adults (such as family members or teachers) to evaluate how this ability manifests in everyday contexts.

In addition, while the linguistic analysis focused on the presence of cognitive and volitional verbs, other relevant linguistic structures, such as complement clauses or causal connectors, were not considered in this study. Further studies could incorporate a more detailed discourse analysis to better understand how children construct mental and emotional explanations. Moreover, to gain a better understanding of the children’s language levels, additional language measures should be employed.

Another limitation lies in the age range of the sample. Although our sample allowed us to observe differences between the DHH and hearing children in their understanding of hidden emotions, we could not observe the entire developmental sequence of this understanding in both groups. Finally, the findings may inform the design of educational interventions focused on promoting the use of cognitive language as a tool to support emotional understanding. In this line, a study conducted in a school context by [10] ([10]) showed that it is possible to improve ToM skills in hearing children through the reading of adapted fairy tales (like “Little Red Riding Hood”) that include sentential complement syntax, causal and contrastive talk about mental states (desires, beliefs, and emotions) and questions that encourage interaction between the children and the teacher. Indeed, a review by [41] ([41]) showed that storytelling may be an effective way to build children’s ToM skills, but just adding metacognitive terms in stories is probably not the best strategy (see [30]). Exploring further the effectiveness of such interventions in inclusive school contexts could provide valuable evidence for the development of pedagogical practices better tailored to the linguistic and socio-cognitive needs of DHH children.

## 5. Conclusions

In conclusion, the present study provides further evidence of a developmental gap in ToM between hearing and non-signing DHH children, particularly in understanding hidden emotions. The most salient points of the study can be summarized as follows:-The hearing children outperformed the DHH children in understanding hidden emotions, both in terms of correct answers and the justifications provided during the task. This result confirms a gap in ToM development between hearing and DHH children.-Although the DHH children used fewer cognitive verbs than the hearing children, in both groups, the use of cognitive verbs (like “think” or “pretend”) was an important predictor of performance for hidden emotion situations. This result highlights the role of cognitive language in complex emotional reasoning.-When age and linguistic skills were taken into account in the analysis, group membership (DHH vs. hearing) was no longer a significant variable. This result suggests that the emotion understanding delay observed in DHH children is related to linguistic factors rather than deafness per se.

## Figures and Tables

**Figure 1 behavsci-15-01106-f001:**
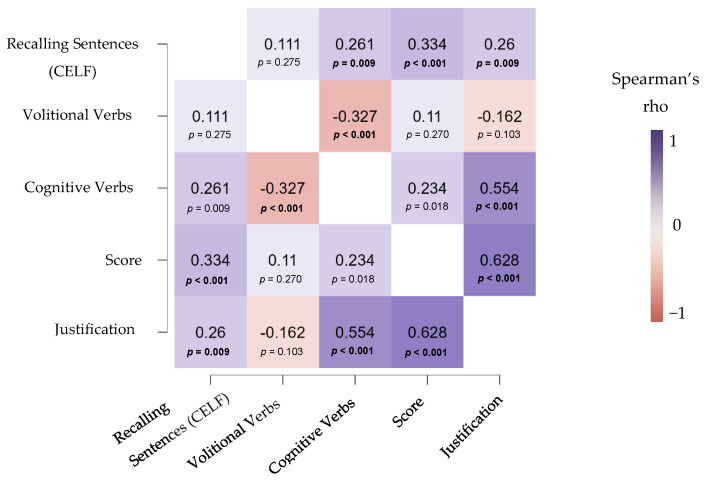
Correlation heatmap between hidden emotion understanding and language.

**Table 1 behavsci-15-01106-t001:** Descriptive characteristics by group (DHH vs. hearing children).

	DHH (*n* = 44)	Hearing (*n* = 59)
Mean age (years)	9.46 (*SD* = 1.17)	9.58 (*SD* = 1.19)
Age range (years)	7.33–12.60	7.01–11.34
Gender (% of girls/% of boys)	43.18%/56.82%	45.76%/54.24%
Age of onset of audiological devices (in months)	38.48 (*SD* = 30.83)	-
Degree of hearing loss	Mild (6.81%) Moderate (43.18%) Severe (15.90%) Profound (34.08%)	No hearing loss
Non-verbal IQ	99.93 (*SD* = 10.10)	101.79 (*SD* = 10.42)

**Table 2 behavsci-15-01106-t002:** Descriptive statistics (means and standard deviations) for the administered tasks.

	CELF	Hidden Emotions		Verbs	
	Recalling Sentences	Score	Justification	Volitional	Cognitive
Range (min.–max.)	(1.000–17.000)	(0–2)	(0–3)	(0–1)	(0–1)
Hearing (*SD*)	9.7586 (3.4555)	1.9153 (0.3845)	2.6780 (0.6549)	0.2203 (0.4180)	0.1395 (0.3506)
DHH (*SD*)	4.4762 (2.8817)	1.7045 (0.5532)	2.2326 (0.8405)	0.6440 (0.4829)	0.3023 (0.4647)

**Table 3 behavsci-15-01106-t003:** Response accuracy by question type (emotion vs. belief).

		Hearing	DHH	Total	
Emotion	Incorrect	3	7	10	*χ*^2^ = 3.369 *p* = 0.066
	Correct	56	37	93	
	Total	59	44	103	
Belief	Incorrect	2	6	8	*χ*^2^ = 3.694 *p* = 0.055
	Correct	57	38	95	
	Total	59	44	103	

**Table 4 behavsci-15-01106-t004:** Use of cognitive and volitional verbs among DHH and hearing children.

		Hearing	DHH	Total	
Cognitive Verbs	Incorrect	21	30	51	*X*^2^ = 11.619 *p* < 0.0001
	Correct	38	13	51	
	Total	59	43	102	
Volitional Verbs	Incorrect	46	37	83	*X*^2^ = 1.0714 *p* = 0.30063
	Correct	13	6	19	
	Total	59	43	102	

**Table 5 behavsci-15-01106-t005:** Model coefficients: hidden emotions.

	Unstandardized Coefficients		Standardized Coefficients	t	Sig.
	B	Desv. Error	Beta		
(Constant)	0.173	0.416		0.415	0.679
CELF (RS)	0.029	0.012	0.232	2.417	0.017
Cognitive Verb	0.278	0.095	0.255	2.936	0.004
Volitional Verb	0.145	0.118	0.102	1.230	0.221
Age	0.011	0.003	0.255	3.221	0.002
Group	0.029	0.060	0.044	0.473	0.637

## Data Availability

The raw data supporting the conclusions of this article will be made available by the authors on request.

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
