# Peer review of "Language and Hidden Emotion Understanding in Deaf and Hard-of-Hearing Children: The Role of Mentalistic Verbs"

_behavsci, 2025, doi:10.3390/bs15081106_

Round 1

Reviewer 1 Report

Comments and Suggestions for Authors

This article is addressing a gap in understanding of hidden emotions for children who are d/hh. The manuscript is well-written and the design of the study is clearly described. I appreciate how the authors dissected the type of language used: cognitive vs. volitional instead of grouping it all together. I believe this study has implications for teacher training in regards to promoting the use of certain language with children who are d/hh to help further the understanding of hidden emotions. This is an important study with findings that are filling a paucity in research. 

I struggle with the connection between children who are d/hh to deaf parents compared to those with hearing parents. The argument is not laid out well enough to be included in this manuscript. In addition, the references from 156-167 are dated when describing outcomes for children who are d/hh. If the authors were to include this piece of the manuscript, it would have been strengthened by having a third group in the study of children who are d/hh from deaf signing families. Because there wasn't his comparison group, the argument of children from homes with hearing families having this delay, is not validated. The same is for 337-339 and 358-360. This seems thrown in and not justified properly. 

I would recommend rewording 174-179. This paragraph needs to be more straight forward and clear on what are the aims and the hypothesis. Be clear and state them directly. 

I would consider reconfiguring some of your tables to make them clearer. For example, I would remove the Total column in Table 1. It is not necessary. I would include more demographic information about the d/hh group in the table. In addition for Table 2, I would suggest having DHH and Hearing as two rows. I would remove minimum and max and instead do a range (min-max). This will provide more clarity. Remember to add captions, labels to the tables. For Table 3 and 5, remind the reader what 0 and 1 stand for. 

Row 223, how were justifications deemed "clear and appropriate?" More information would be helpful.

Who conducted the sessions? What is their training? 

Why was that specific subtest used for understanding of language levels? That is one small part of the CELF. If you wanted to get a grander understanding of their language levels, a more complete CELF would be helpful. 

Author Response

Comment 1: I struggle with the connection between children who are d/hh to deaf parents compared to those with hearing parents. The argument is not laid out well enough to be included in this manuscript. In addition, the references from 156-167 are dated when describing outcomes for children who are d/hh. If the authors were to include this piece of the manuscript, it would have been strengthened by having a third group in the study of children who are d/hh from deaf signing families. Because there wasn't his comparison group, the argument of children from homes with hearing families having this delay, is not validated. The same is for 337-339 and 358-360. This seems thrown in and not justified properly. 

Response 1: Thank you for your comment; it helped us update the references of this part. Also, we have now clarified better in the introduction the differences in Theory of Mind (ToM) performance between deaf children born to deaf signing parents and those born to hearing parents. The literature consistently shows that ToM delays are primarily observed in the latter group. While our study does not focus on deaf children of deaf parents, we agree with Reviewer 2 that the sign language literature cannot be overlooked in this context. Therefore, we have included relevant references and discussions to acknowledge its importance. (lines 153-162)

Comment 2: I would recommend rewording 174-179. This paragraph needs to be more straight forward and clear on what are the aims and the hypothesis. Be clear and state them directly. 

Response 2: Thank you for your helpful suggestion. We agree that the original paragraph lacked clarity. We have now rewritten it (lines 187- 193 & 201–206) to state the aims and hypotheses more explicitly and directly.

Comment 3: I would consider reconfiguring some of your tables to make them clearer. For example, I would remove the Total column in Table 1. It is not necessary. I would include more demographic information about the d/hh group in the table.

Response 3: We agree that the format of some of the tables included in the article is unclear. Regarding Table 1, and following the recommendation of Reviewer 1, we have removed the last column (“Total”) to make the table clearer for readers. Although we appreciate Reviewer 1’s suggestion to include more demographic information about the DHH participants, we believe that the information currently included in the table is sufficient and appropriate given the objectives of the study. We believe that including additional demographic information about the DHH participants that is not relevant to the study’s objectives or not discussed in the analysis could potentially lead to confusion for the reader.

Comment 4: In addition for Table 2, I would suggest having DHH and Hearing as two rows. I would remove minimum and max and instead do a range (min-max). This will provide more clarity. Remember to add captions, labels to the tables. For Table 3 and 5, remind the reader what 0 and 1 stand for.

Response 4: Following the recommendation of Reviewer 1, Table 2 has been reorganized by placing the study groups (hearing and DHH) in two separate rows. The minimum and maximum values for each group have been removed, and a new row has been added to indicate the combined range for both groups. Acronyms have been removed from the table and replaced with their corresponding full expressions. The table title was also revised as a result of these changes. In Table 3 and Table 5 (now Table 4), the values 0 and 1 have been replaced with the expressions “incorrect” and “correct,” respectively. 

Comment 5: Row 223, how were justifications deemed "clear and appropriate?" More information would be helpful.

Response 5: Thank you for your comment. We agree that additional clarification was needed regarding the criteria for scoring justifications as “clear and appropriate.” We have now added this sentence (lines 258-259): “We considered a justification about the external emotion as correct if it referred to the fact that the child was lying or trying to deceive her parents”.

Comment 6: Who conducted the sessions? What is their training? 

Response 6: Thank you for your observation. We have now clarified this point in the manuscript (lines 300–302). Specifically, we added the following sentence: “The sessions were conducted by researchers with prior experience administering language and Theory of Mind tasks with both DHH and hearing children.” This addition highlights the training and background of the assessors, ensuring the reader that the sessions were conducted by appropriately qualified personnel.

Comment 7: Why was that specific subtest used for understanding of language levels? That is one small part of the CELF. If you wanted to get a grander understanding of their language levels, a more complete CELF would be helpful.

Response 7: Thank you for this insightful comment. We agree that a more comprehensive language assessment, such as the full CELF battery, could provide a broader understanding of the children's language abilities. However, our primary focus was on Theory of Mind (ToM) performance, particularly on hidden emotions and the reasoning involved in justifying responses. For this reason, we selected the Recalling Sentences subtest of the CELF, as it includes a variety of syntactic structures and goes beyond vocabulary tasks. This subtest was deemed most relevant for capturing aspects of language that directly support ToM reasoning. Given the wider scope of the project (which also involves executive function tasks that are cognitively demanding) we made the methodological decision to limit the number of assessments in order to avoid fatigue, especially among DHH children. We have acknowledged this limitation in the manuscript (lines 473–474): “Moreover, to gain a better understanding of the children’s language level, additional language measures should be employed.”

Reviewer 2 Report

Comments and Suggestions for Authors

Review of the manuscript "Language and Hidden Emotion Understanding in Deaf and Hard of Hearing Children: The role of Mentalistic Verbs"
The paper examines the connection between language and hidden emotions in deaf and hard of hearing children, through examining the use of cognitive and volitional verbs and theory of mind.
The paper provides an adequate overview of the concepts that the research deals with.
In the Materials and Methods section, in the description of Participants, it was not stated whether the groups were equalized according to gender. It would be useful to have that data, and in case there is a difference, to take it into account in the interpretation of the data.
Instruments and procedures are adequately presented.
As part of statistical data processing, the authors note that non-parametric statistical analyzes were used, and it would be good to state the reasons for this. Given that multiple linear regression is coming after the previous analyses, it is necessary to indicate whether the prerequisites required by it have been tested and met. Further presentation of results, discussion and conclusions depend on this fact.
In the Results section, for the sake of clarity, it would be preferable if the tables did not go one after the other, but there should be text between them. Likewise, each table should be accompanied by an appropriate legend.
Clarity is also reduced by the display of results, where different parameters are rounded to different number of decimal places.
By analyzing the literature used, only three references from the last 5 years are found, so it is recommended to supplement the literature with more recent references (especially through discussion).
Abstract - correct according to valid results
Keywords - avoid words that are already in the title of the manuscript

Author Response

Comment 1: In the Materials and Methods section, in the description of Participants, it was not stated whether the groups were equalized according to gender. It would be useful to have that data, and in case there is a difference, to take it into account in the interpretation of the data.

Response 1: Thank you for pointing this out. We have now included the gender distribution for each group in Table 1 to provide a clearer picture of the sample composition. 

Comment 2: As part of statistical data processing, the authors note that non-parametric statistical analyzes were used, and it would be good to state the reasons for this. Given that multiple linear regression is coming after the previous analyses, it is necessary to indicate whether the prerequisites required by it have been tested and met. 

Response 2: Thank you for this important observation. We have now clarified in the manuscript the rationale behind our use of non-parametric statistical analyses. Specifically, non-parametric tests were employed because several variables in our dataset violated the assumptions of normality, as assessed by the Shapiro–Wilk test. This choice was made to ensure more robust and reliable results when comparing groups or examining associations in variables with skewed distributions. Regarding the multiple linear regression, we have now added a statement confirming that the necessary assumptions were tested and met. In particular, we verified the normality of residuals, linearity, homoscedasticity, and absence of multicollinearity (lines 358-362).

Comment 3: In the Results section, for the sake of clarity, it would be preferable if the tables did not go one after the other, but there should be text between them. Likewise, each table should be accompanied by an appropriate legend.
Clarity is also reduced by the display of results, where different parameters are rounded to different number of decimal places.

Response 3: Thank you for these valuable suggestions regarding the clarity and formatting of the Results section. We have revised the manuscript accordingly. We have added explanatory text between the tables to guide the reader through the results and ensure a smoother flow of information. Each table now includes either a clear and descriptive legend or the variable labels are directly incorporated within the table. We have standardized the rounding of all numerical values across tables and text, ensuring that statistical parameters are consistently reported with the same number of decimal places for improved readability and coherence.

Comment 4: By analyzing the literature used, only three references from the last 5 years are found, so it is recommended to supplement the literature with more recent references (especially through discussion).

Response 4: We thank the reviewer for this valuable recommendation, which has contributed to strengthening both the Introduction and the Discussion sections. We have now incorporated three more recent and relevant references:

Chu (2025) has been cited in the Introduction (lines 173) and again in the Discussion (lines 426-430), as it provides recent insights into Theory of Mind development in hearing-impaired children with varying degrees of hearing loss.

Vachio et al. (2023) has been included in the Introduction (lines 196–200) and the Discussion (lines 444), supporting our interpretation regarding the use of mental state verbs and complex syntax in DHH children.

Meristo & Surian (2025) has included in the Discussion (line 397).

Other references of the last 5 years that we have now included are: Yu et al. (2021), Peters & Pisoni (2023), Peterson, (2021), Richardson et al., 2020

Comment 5: Keywords - avoid words that are already in the title of the manuscript.

Response 5: Thank you for this helpful observation. We have revised the list of keywords to avoid repetition of terms already included in the title. The updated keywords better reflect the conceptual scope of the study and improve its discoverability across related domains.

Keywords: Social cognition, Belief attribution, Mental state reasoning, Childhood deafness, Language development.

Reviewer 3 Report

Comments and Suggestions for Authors

Please see attached comments

Author Response

Overall

Comment 1: In a few places, the authors use deficit-focused language, like “hearing deficit”, “hearing impairment”. The English-speaking deaf community typically prefers more neutral language, that doesn’t frame deafness as an impairment. The authors could replace these terms with “deafness”

Response 1: We apologise for the use of non-adequate terms. We have searched the document for references to “hearing deficit” or “hearing impairment” and have replaced them with more appropriate terms or expressions.

Comment 2: The authors allude to the role of language access but seem to ignore the sign language literature on this topic. Some papers that might be helpful:
- Richardson et al., 2020: https://www.nature.com/articles/s41467-020-17004-y

Response 2: Thank you for this recommendation. We have included some information related to the non-difficulties in ToM in native deaf signers both in the introduction and discussion. In the introduction, we wrote: In this regard, however, some researchers have suggested that the ability of parents to communicate with children through a sign language from birth, as is the case with deaf children born to deaf parents (DoD children), is crucial for the development of socio-cognitive skills (Gagne et al., 2019; Meristo et al., 2007; Richardson et al., 2020). There is long evidence nowadays that DoD children outperform DoH children; either oral communicators or late signers) in ToM tasks, and  that DoD children usually perform similarly to hearing peers of their same age (see Peterson, 2021) (lines 156-162). In the discussion, we wrote: “In this regard, we need to highlight that our results do not apply to deaf signers, DoD children who have sign language as their native language, as the literature suggests that unless they have a limitation on the use of their sign language in their environment, their performance in ToM tasks is similar to that of their hearing peers (Peterson, 2021)” (lines 400-403).

Abstract

Comments 3:

  • Hidden emotions should be defined briefly the first time it’s mentioned - “The hidden emotions task presented children with an illustrated story in which a character pretends to feel a certain emotion in front of two observers, only one of whom knowing the real feelings of the child.” – It’s unclear what’s actually being measured.
  • “The use of cognitive verbs predicted performance across groups…” The use of cognitive verbs by whom?
  • When describing the participant population, authors should specify the language used by the children. 

Response 3: We thank the reviewer for these suggestions, which have helped improve the clarity and precision of our manuscript. We have revised the entire abstract accordingly.

Intro
Comment 4: The Introduction requires substantial reworking. It reads like an instructive text, where the authors are summarizing their knowledge of a topic rather than engaging more deeply with the state of the literature. Rather than giving a broad overview of theory of mind, the authors should focus in on why hidden emotion understanding is compelling and merits research.

Response 4: We thank the reviewer for this important comment. We agree that the original version of the Introduction focused too broadly on Theory of Mind and did not sufficiently frame the specific significance of hidden emotion understanding as a research focus. In response, we have substantially revised the Introduction to provide a more targeted and conceptually engaged rationale. Rather than offering a general overview of ToM, the revised section now emphasizes: Why hidden emotion understanding is a particularly compelling and developmentally relevant aspect of ToM; How this ability draws on both emotional inference and mental state attribution; Why it may be especially vulnerable in populations with delayed or limited language access, such as deaf and hard of hearing (DHH) children and how previous work supports the need for targeted research in this area.

Comment 5:  Here and in the Discussion, more citations are needed. Several sentences stood out as needing a reference, but none were provided:
a) “ In this regard, some researchers have suggested that the ability of parents to communicate with children through a sign language from birth (as is the case with deaf children born to deaf parents) is crucial for the development of socio-cognitive skills (among them the understanding of emotions).”
b) “Several investigations suggest that DHH children born to hearing non-signing parents may have difficulties with ToM, that is, understanding the mental states of others, such as desires, beliefs or intentions, and in particular emotions (Peterson et al., 2016).” –this sentence says “several studies” but then only one reference is listed.

Response 5: We thank the reviewer for this helpful observation. We agree that the original version lacked sufficient referencing in key parts of both the Introduction and Discussion. We have now revised the text to include appropriate citations that support the claims made.

a) "In this regard, however, some researchers have suggested that the ability of parents to communicate with children through a sign language from birth, as is the case with deaf children born to deaf parents (DoD children), is crucial for the development of socio-cognitive skills (Gagne et al., 2019; Meristo et al., 2007; Richardson et al., 2020) (lines 156-159).

b)"Several investigations suggest that DHH children born to hearing non-signing parents (DoH children) may show differences in ToM development, compared to their hearing peers, that is, in understanding the mental states of others, such as desires, beliefs or intentions, and in particular emotions (e.g. Peterson et al., 2016; Schick et al., 2007)" (lines 146-150).

Method
Comment 6: Has the CELF sentence repetition task been validated for use with DHH children?

Response 6: Despite the manual of the CELF-V does not provide a validation of the test for DHH children, in the study by Durán-Bouza et al. (2024), where they evaluated hearing and deaf children with the CELF-V, they found that this test showed high internal consistency in the deaf group for the different subtests administered to the children, including the recalling sentences subtest.

Durán-Bouza, M., Pernas, L., & Brenlla-Blanco, J. C. (2024). Assessment of Linguistic Profile of Oral-Language-Proficient Hearing-Impaired Children Using Clinical Evaluation of Language Fundamentals: (CELF5). Children, 11(12), 1458.

Comment 7: I was surprised that the analysis did not differentiate in any way (that I could tell) between points awarded for answering the control questions and points awarded for identifying the hidden emotion. In this way, it doesn’t seem like the “hidden emotion” score necessarily reflects hidden emotion understanding. E.g., Child A could answer the control questions wrong and the other questions correctly and score a 2. Child B could answer the hidden emotion question wrong but get the other questions correct and also score a 2. Child A and B would have identical scores. 

Response 7: We realized that we had not been enough accurate in describing how the Hidden emotions task had been scored. In fact, no points were awarded for the control questions. We explain it now with the following sentences “The majority of children 86.4 % responded correctly to all memory questions, while 11.7% failed one question and only 2 children (one from each group) failed 2 questions. However, we did not exclude for their performance in the memory questions as we prioritised their performance in the understanding of the emotions and beliefs of the characters.” So, we gave 1 point for understanding the emotions (external and internal) of the character, one point for understanding the beliefs of the parents, and one points for justifying correctly the external emotion of the character. This explanation reads now as follows: “The scoring of the task was as follows: children were awarded 1 point for correctly answering both the external and internal emotion questions. A second point was given for accurately identifying the beliefs of the observers—specifically, selecting the protagonist’s internal emotion for the parent who witnessed the previous situation, and the protagonist’s external emotion for the parent who did not witness it. Finally, a third point was awarded when the child provided a clear and appropriate justification for their response to why the protagonist showed the external emotion the child had said”.

Results

Comment 8: The statistical approach is generally appropriate for the data - “Language ability (RS_CELF) showed a marginal effect (β = .0140, p = .0028),” the p value reported here is different from the p value reported in the table.

Response 8: Thank you for catching this discrepancy. You are absolutely right—the p-value reported in the text did not match the value shown in the corresponding results table. We have carefully reviewed the table and regression output and corrected the inconsistency.

Comment 9: Minor comment: the subheadings under Results (Preliminary Analysis, Main Analysis) are uninformative.

Response 9: Thank you for this helpful comment. We agree that the subheadings “Preliminary Analysis” and “Main Analysis” were too generic and did not add meaningful structure to the Results section. In response, we have removed these subheadings entirely to improve the flow and clarity of the text. The results are now presented in a continuous, logically ordered narrative without unnecessary divisions.

Comment 10: For the regression, some measure of model fit is needed. I also recommend that the authors conduct a vif test to check whether multicollinearity could be affecting their statistical results.

Response 10: Thank you for this valuable comment. In response, we have now added a measure of model fit to the Results section (line 374). Furthermore, following your recommendation, we conducted VIF analysis to assess multicollinearity among the predictors. All VIF values were below the commonly accepted threshold of 2, indicating that multicollinearity is not a concern in our model (line 360-361).

Comment 11: This would need to be an exploratory analysis, but these results make me curious about interactions between the linguistic variables and task performance. E.g., is language a stronger predictor of performance for the DHH children? Is age a weaker predictor for DHH children?

Response 11: We thank the reviewer for this thoughtful suggestion. We find  that exploring potential interactions between linguistic predictors and group membership (DHH vs. hearing) would be very interesting for our study. However, we believe that our sample is too small for carrying out this analysis adequately.

Discussion
Comment 12: Discussion needs more theory. Why is this interesting? What does this suggest?

Response 12: We tried to improve the discussion in several ways. First, in line with the commentary of the reviewer, we tried to explain why this study is interesting. We have added the following texts to the discussion:

We tried to highlight why the new task of hidden emotions that we created can be interesting for future research: “This variation allows us to assess adequately the understanding of the beliefs of the observers in hidden emotion situations, which will be helpful for research in this topic in future studies.” (line 410-412).

We tried to describe how the understanding of hidden emotions could be fostered in educational settings. We said now: “Our results are interesting because they focus on an area of socio-cognitive development that has been understudied in the last years (whether children realize that hiding emotions may have consequences on the belief of certain observers as a function of the knowledge of the observers about the situation), and by showing that some DHH children may have difficulty in this skill, educators should make sure that children can reason adequately in this kind of situations, and help them develop and apply this knowledge. One approach to do so could be by using stories of hidden emotions in which the beliefs of the stories are depicted as thought bubbles, as Wellman and Peterson (2013) found that deaf children’s ToM understanding can improve by having conversations about false beliefs using thought bubbles and mental-state terms.” (lines 421-431).

In a similar vein, we added another text in another paragraph in which we said “The limited use of cognitive terms observed in DHH non-signing children in our study when giving explanations about the emotional expression of the character when it was hiding his real emotion could be perhaps fostered by having cognition-oriented conversations that include mental-state terms (like think, know, intend or pretend) and where children are asked about the thoughts and knowledge of the characters and of themselves (Peterson, 2020).”

Tables
Comment 13: Table 1: Gender is under-documented. Rather than just reporting n for female, authors should report all levels. Because the groups (DHH vs. hearing) differ in N, percentages would be more useful to the reader than raw counts. 

Response 13: We agree that, given the different group sizes, it is more appropriate to report percentages rather than raw n values. We also acknowledge the relevance of including percentages for all gender categories considered (women and men). Taking these two aspects into account, in Table 1, the n values have been replaced with percentages, and both the percentage of girls and boys have been indicated for each group. 

Furthermore, we made an statistical comparison through the Chi-Square to make sure that gender was not different for boys than for girls, and we wrote the following paragraph in the participants section: “Furthermore, the Chi-Square test showed that the variables group and gender were not related (χ² = .068; p = 0.79), as the proportion of boys and girls in both groups were very similar.”

 Comment 14: Table 2: I recommend converting this table into Figures, perhaps a set of violin plots. Currently, it just reads as a wall of numbers, and it is unclear what the reader should focus on.

Response 14: We appreciate Reviewer 3’s comment regarding the possibility of converting Table 2 into a set of figures, such as violin plots. However, we believe that keeping the table format—after incorporating the suggestions made by Reviewer 1—is more appropriate in this case, for two main reasons. First, the table allows us to present the exact values for each variable, which is not possible with a figure, where the focus is often on overall visualization rather than numerical detail. We consider that figures are more suitable when the goal is to illustrate general trends or distributions. Second, we have revised and reformulated Table 2 to make it clearer and more accessible; it now presents the information in a more structured and readable way, avoiding the “wall of numbers” effect.

Comment 15: I strongly encourage the authors to minimize their use of acronyms (HE_Just, etc.) in the tables. 

Response 15: In Table 2, the acronyms have been replaced with their corresponding full expressions. The notes originally included in this table have been removed. Table 4 has been converted into Figure 1 as a result of Comment 4 from Reviewer 2. In Figure 1, the acronyms have been replaced with their corresponding full expressions.

Comment 16: Table 3: Correct/Incorrect would be easier for readers to quickly understand than 1/0.

Response 16: In Table 3 and Table 5, the values 0 and 1 have been replaced with the expressions “incorrect” and “correct,” respectively. 

Comment 17: Table 4 could also be replaced with a figure. Correlation heatmaps can be very effective for reporting this type of analysis/result.

Response 17: We greatly appreciate Reviewer 2’s comment regarding Table 4. Following their recommendations, Table 4 has been converted into Figure 1, which now displays a heatmap correlation matrix.

Round 2

Reviewer 2 Report

Comments and Suggestions for Authors

The authors corrected the manuscript in accordance with the suggestions.
The discussion in the text has been greatly improved. The only thing that should be considered is how to present the data in the table with correlations as clearly as possible, especially since the level of significance is not presented.

Author Response

Comment 1:  The only thing that should be considered is how to present the data in the table with correlations as clearly as possible, especially since the level of significance is not presented.

Response 1: Thank you very much for your comment. We have addressed this by adding significance markers to Figure 1, which now clearly indicates the level of significance for each correlation. We hope this improves clarity and interpretability of the results.

Reviewer 3 Report

Comments and Suggestions for Authors

Abstract

  • The authors adequately addressed all of my comments

Intro

  • The introduction is much improved, but I have some additional comments. 
    • “With the emergence of language, during the first to years of life, children start to label basic emotions (happiness, sadness…) based on facial expressions, movements and voice. During their second and third years of life, children start to understand which situations cause certain emotions, and also that people’s emotions depend on their desires.” – citations needed for each of these statements
    • Line 167: the word “children” missing after DoH
    • “Therefore, the emotional understanding skills of deaf children must be framed in more general skills that they have when it comes to understanding people's mental states and behaviors.” – this sentence was difficult to parse. What are the authors trying to say here?
    • This sentence is overly long and should be split up. “Hence an important factor in the development of children's ability to understand mental and emotional states is access to a first language as soon as possible, whether oral or signed, but also the fact of being able to use it to talk about the thoughts and feelings of others, both at home and at school (Meristo et al. 2007).”

Methods

  • Has the CELF sentence repetition task been validated for use with DHH children?
    • I recommend that the authors include this information (briefly) in the Methods section.
    • “Despite the manual of the CELF-V does not provide a validation of the test for DHH children, in the study by Durán-Bouza et al. (2024), where they evaluated hearing and deaf children with the CELF-V, they found that this test showed high internal consistency in the deaf group for the different subtests administered to the children, including the recalling sentences subtest.”
  • “Furthermore, the Chi-Square test showed that the variables group and gender were not related (χ² = .068; p = 0.79), as the proportion of boys and girls in both groups were very similar.” - the Chi-Square test cannot determine that group and gender are not related. Instead, the authors could say something like: “Furthermore, the Chi-Square test showed that the gender distribution did not differ significantly by group; the proportions of boys and girls in each group were quite similar.”

Results 

  • The authors adequately addressed my comments.
  • Paragraph starting at line 335: I strongly recommend removing the acronyms from the text. Even your most dedicated readers will not remember all of these, and would need to flip back and forth to find what the acronyms mean.

Figures/Tables

  • I appreciate that the authors converted Table 4 into Figure 1. I would just recommend switching the acronyms from “C_Verbs” to “Cognitive Verbs”, etc., as they have already helpfully done elsewhere.
  • Table 5: minimize use of acronyms for your predictors

Abbreviations (line 510):

  • DoH, DoD, and CELF need to be added to this section.

Author Response

Introduction

Comment 1: "With the emergence of language, during the first to years of life, children start to label basic emotions (happiness, sadness…) based on facial expressions, movements and voice. During their second and third years of life, children start to understand which situations cause certain emotions, and also that people’s emotions depend on their desires.” – citations needed for each of these statements. 

Response 1: Thank you for this helpful comment. Apologies for the confusion — all the information in that sentence comes from the same source (Pons & Harris, 2019), and we initially avoided repeating the citation to maintain readability. However, you're right that this can lead to ambiguity. We have now repeated the citation

Methods

Comment 2: Has the CELF sentence repetition task been validated for use with DHH children? I recommend that the authors include this information (briefly) in the Methods section. “Despite the manual of the CELF-V does not provide a validation of the test for DHH children, in the study by Durán-Bouza et al. (2024), where they evaluated hearing and deaf children with the CELF-V, they found that this test showed high internal consistency in the deaf group for the different subtests administered to the children, including the recalling sentences subtest.

Response 2: Thank you for this relevant observation. Although the CELF-V manual does not include specific validation data for DHH children, we have now added a sentence in the Methods section (lines 275–279) clarifying that, in the study by Durán-Bouza et al. (2024), high internal consistency was observed in the DHH group across several CELF-V subtests, including the Recalling Sentences subtest.

Comment 3: “Furthermore, the Chi-Square test showed that the variables group and gender were not related (χ² = .068; p = 0.79), as the proportion of boys and girls in both groups were very similar.” - the Chi-Square test cannot determine that group and gender are not related. Instead, the authors could say something like: “Furthermore, the Chi-Square test showed that the gender distribution did not differ significantly by group; the proportions of boys and girls in each group were quite similar.”

Response 3: Thank you for pointing this out. We have now revised the sentence and incorporated the suggested phrasing: “The Chi-Square test showed that the gender distribution did not differ significantly by group; the proportions of boys and girls in each group were quite similar.” 

Results 

Comment 4: Paragraph starting at line 335: I strongly recommend removing the acronyms from the text. Even your most dedicated readers will not remember all of these, and would need to flip back and forth to find what the acronyms mean.

Response 4: Thank you for the suggestion. We have removed the acronyms from this paragraph and replaced them with the full terms to improve readability.

Figures/Tables

Comment 5: I appreciate that the authors converted Table 4 into Figure 1. I would just recommend switching the acronyms from “C_Verbs” to “Cognitive Verbs”, etc., as they have already helpfully done elsewhere.

Response 5: Thank you for the helpful suggestion. We have updated the labels in Figure 1 accordingly, replacing all acronyms with their full forms.  Additionally, following a comment from Reviewer 2, we have also added significance markers to the figure to facilitate interpretation of the results.

Comment 6: Table 5: minimize use of acronyms for your predictors.

Response 6: We have revised Table 5 to reduce the use of acronyms and improve clarity.

Abbreviations (line 510):

Comment 7: DoH, DoD, and CELF need to be added to this section.

Response 7: These abbreviations have now been added to the list at the end of the manuscript.